# Single-Nucleus RNA-Seq: Open the Era of Great Navigation for FFPE Tissue

**DOI:** 10.3390/ijms241813744

**Published:** 2023-09-06

**Authors:** Yunxia Guo, Wenjia Wang, Kaiqiang Ye, Liyong He, Qinyu Ge, Yan Huang, Xiangwei Zhao

**Affiliations:** State Key Laboratory of Digital Medical Engineering, School of Biological Science & Medical Engineering, Southeast University, Nanjing 210096, China; 15150517535@163.com (Y.G.); 18800113229@163.com (W.W.); kaiqiangye1104@163.com (K.Y.); heliyong@seu.edu.cn (L.H.); geqinyu@seu.edu.cn (Q.G.); hylucky@seu.edu.cn (Y.H.)

**Keywords:** FFPE, snRNA-seq, nuclear preparation strategies

## Abstract

Single-cell sequencing (scRNA-seq) has revolutionized our ability to explore heterogeneity and genetic variations at the single-cell level, opening up new avenues for understanding disease mechanisms and cell–cell interactions. Single-nucleus RNA-sequencing (snRNA-seq) is emerging as a promising solution to scRNA-seq due to its reduced ionized transcription bias and compatibility with richer samples. This approach will provide an exciting opportunity for in-depth exploration of billions of formalin-fixed paraffin-embedded (FFPE) tissues. Recent advancements in single-cell/nucleus gene expression workflows tailored for FFPE tissues have demonstrated their feasibility and provided crucial guidance for future studies utilizing FFPE specimens. In this review, we provide a broad overview of the nuclear preparation strategies, the latest technologies of snRNA-seq applicable to FFPE samples. Finally, the limitations and potential technical developments of snRNA-seq in FFPE samples are summarized. The development of snRNA-seq technologies for FFPE samples will lay a foundation for transcriptomic studies of valuable samples in clinical medicine and human sample banks.

## 1. Introduction

Detecting gene expression in individual cells can identify cell type and cell state. Since Tang developed single-cell RNA sequencing (scRNA-seq) in 2009 (ref mRNA-Seq whole-transcriptome analysis of a single cell), it has replaced bulk RNA-sq as a powerful tool for studying cell transcription profiles [1]. However, the cell dissociation methods required for scRNA-seq lead to experimental changes in gene expression and cell death. In addition, many valuable samples cannot be obtained as fresh tissue and frozen samples, thus limiting research on archival and biobank materials. Single-nucleus RNA sequencing (snRNA-seq) becomes an attractive alternative to scRNA-seq by analyzing the nucleus rather than the whole cell, allowing the study of frozen or hard-to-dissociate tissues due to accurate identification of cell types, reducing dissociation-induced transcription to discover unknown and rare cell subpopulations. In addition, snRNA-seq reduces bias in cell coverage and can be applied to archived frozen specimens [2], and is less susceptible to perturbations in gene expression that occur during cell separation, such as increased expression of direct early genes that mask transcriptional signatures of neuronal activity [3].

Although fresh or fresh frozen clinical samples are ideal for transcriptomic analysis, the limited availability of these samples is a serious drawback. Formalin-fixed paraffin-embedded (FFPE) tissue blocks represent the paramount approach for preserving human tissue in clinical diagnostics [4]. Globally, pathology laboratories and sample banks house over a billion FFPE sections. These repositories offer a precious resource for profound transcriptomic analysis, albeit accessible only weeks to years post sample collection. This delay is necessitated by critical clinical attributes, such as tumor genetics, treatment response, and patient survival, which require adequate time for meaningful development [5]. The compatibility of sc/snRNA-seq with FFPE samples allows researchers to study all aspects of tissue cell heterogeneity [6,7]. Recently, sc/snRNA-seq chemistry has been developed for FFPE single-cell/nucleus suspensions, and the results have shown that this allows performing sc/snRNA-seq of nuclei isolated from FFPE tissue samples [8,9,10,11,12] at read depths that allow a similarly fine-grained analysis compared to fresh or frozen cell suspensions.

However, RNA degradation in FFPE samples is severe compared to DNA due to the chemical modification [13], poly(A) tail damage [14], and covalent modification of RNA nucleotide bases by monomethylol addition. These covalent modifications can impact reverse transcription from mRNA to cDNA and significantly alter gene expression profiling [15]. Therefore, it is necessary to carefully consider the pre-treatment of FFPE samples and appropriate enzymatic hydrolysis methods. In addition, snRNA-seq techniques for FFPE samples have not been reviewed. In this review, we provide a broad overview of the nuclear preparation strategies, the latest technologies of snRNA-seq applicable to FFPE samples. Finally, the limitations and potential technical developments of snRNA-seq in FFPE samples are summarized.

## 2. Isolation Strategies and Applications of Nuclei from FFPE Tissues

Accurate transcriptomic characterization of each cell in clinical FFPE specimens is believed to provide a better understanding of cell heterogeneity and population dynamics, thereby improving accurate diagnosis, treatment, and prognosis of human disease. The application of sc/snRNA-seq in FFPE samples is premised on obtaining superior single-cell or single-nucleus suspensions. However, isolation of intact single cells/nuclei remains challenging due to RNA crosslinking, modification, and degradation caused by formaldehyde fixation. Although there is a commercially available kit for FFPE samples (Miltenyi Biotech FFPE Tissues Dissociation Kit), it is highly recommended to perform snRNA-seq instead of scRNA-seq on FFPE tissues, since the fixation process often damages the integrity of various cellular structures, leading to the detection of heavily degraded cytoplasmic RNA [16]. The preparation methods of FFPE sample nucleus are longstanding and are mainly divided into two categories: (I) enzymatic dissociation strategies and (II) mechanical extraction strategies.

### 2.1. Enzymatic Dissociation Strategies

The nuclear dissociation methods for FFPE tissues date back to the last century (Table 1). These protocols share a common procedure of dewaxing and rehydration of FFPE slides, but they differ in terms of the type or concentration of digestive enzymes used and reaction conditions employed. Hedley et al. first described the release of nuclei from archived tissue in 1983 [17], and measured the cellular DNA content of FFPE human tumors using flow cytometry (FCM). In detail, FFPE sections were incubated in a pepsin dissociation solution (0.5% pepsin, 0.9% NaCl, 2N HCl) at 37 °C for 30 min to prepare a nuclei suspension. Since then, this protocol [17] has undergone several modifications and improvements to be suitable for paraffin-embedded materials, and applied to in situ hybridization (ISH) of DNA probes [18], as well as fluorescence in situ hybridization (FISH) [19,20,21,22]. In 1985, Schutte et al. [23] dissociated the nuclei of FFPE tumor samples using a trypsin solution and determined the DNA ploidy levels by FCM. The samples were incubated overnight at 37 °C in 0.25% trypsin buffer (3 mM trisodium citrate, 0.1% Nonidet P40, 1.5 mM spermine tetrachloride, 0.5 mM Tris). In addition, proteinase K is also commonly used to dissociate the nuclei of FFPE samples. Liehr et al. [24] proposed covering FFPE sections on a slide with protease K buffer (5 mg protease K, 1 M Tris (pH 7.5), 0.5 M EDTA (pH 7.0), 5 M NaCl). This method allowed for nucleus extraction from a single mounted 5 μm section, and was applied in FISH tests for head and neck squamous cell carcinomas (HNSCCs).

Various methods exist for the isolation of FFPE nuclei, with the majority being employed for FISH. In 2012, Michael et al. [25] applied flow cytometry-based methods to isolate pure populations of tumor cell nuclei from FFPE tissues. They developed a methodology compatible with oligonucleotide array comparative genomic hybridization (aCGH) and whole-exome sequencing (WES) analyses. Specifically, FFPE scrolls were de-crosslinked by heating in an EDTA solution, followed by washing with CaCl_2_; then, overnight digestion in a mixed dissociation solution of multiple enzymes (collagenase type 3, purified collagenase, and hyaluronidase) was performed. The resulting pellets were then passed through a 25 G needle approximately 10–20 times. In 2017, Hicks et al. [26] followed this protocol with slight modifications [25]. They introduced and validated an effective approach to perform single-nuclei whole-genome copy number profiling for FFPE clinical samples, overcoming the limitation of single-nuclei genomics, which had been restricted to the analysis of fresh or frozen tissue. 

**Table 1 ijms-24-13744-t001:** Enzymatic dissociation strategies and applications of nuclei from FFPE tissues.

Tissues	Thickness	Protease	Time	Application	References
tumor	30 μm	0.5% pepsin	30 min	DNA content	[17]
brain, lung, breast, testis, kidney, and colon tumor	50 μm	0.5% pepsin	30 min	ISH	[18]
——	50 μm	0.5% pepsin	30 min	FISH	[19]
brain	——	0.5% pepsin	30 min	FISH	[21]
breast, ovary, tumor	40 μm	0.005% pepsin	2 h	FISH	[22]
tumor	——	0.25% trypsin	Overnight	DNA ploidy	[23]
——	20–30 μm	0.005% PK	30 min	FISH	[20]
HNSCC	5 μm	PK	1 h	FISH	[24]
lymph nodes, tonsils	——	0.01% PK	2 h	ISH, FISH	[27]
liver cancer	10 μm	0.005% PK	2 h	FISH	[28]
breast, brain, bladder, ovarian, and pancreas	40–60 μm	collagenase, hyaluronidase	Overnight	aCGH, WES	[25]
breast cancer	100 μm		16 h	sn-WGSA	[26]

PK: proteinase K; ISH: in situ hybridization; FISH: fluorescence in situ hybridization; aCGH: array comparative genomic hybridization; WES: whole-exome sequencing; sn-WGSA: single-nucleus whole-genome DNA copy number; HNSCC: head and neck squamous cell carcinoma.

### 2.2. Mechanical Extraction Strategies

Recently, Regev et al. extracted nuclei from FFPE samples using the traditional method for frozen samples in the snFFPE-Seq technique [10]. They first developed a protocol to obtain intact nucleus suspensions from FFPE samples of mouse brain by optimizing the deparaffinization and rehydration process. In detail, they worked with 50 µm scrolls of the cortex area, and three deparaffinization treatments were tested: mineral oil with heat (80 °C), xylene with heat (90 °C), and xylene at room temperature. Nuclei extraction was then performed using a previously developed lysis buffer that preserved the attachment of ribosomes to the nuclear membrane, thereby increasing the yield of captured RNA molecules. At the same time, Martelotto et al. conducted a study involving the combination of enzymatic dissociation and mechanical extraction to isolate nuclei from FFPE breast cancer metastases to the liver [9]. They followed the nuclear dissociation protocol developed by Michael et al. to treat 25 μm FFPE slices [25]. Subsequently, the tissue was homogenized in Ez lysis buffer, following the same steps as the nucleation process used for frozen samples. Similarly, Wang et al. [12] proposed a mechanical-binding enzyme dissociation method to extract the nucleus from FFPE samples, and applied snRNA-seq to a clinical FFPE specimen of human liver cancer. The FFPE slices were dewaxed and hydrated, homogenized in precooled 0.1% Triton X-100 buffer, and then dissociated in 10 mg/mL proteinase K buffer solution. Finally, the isolated nuclei were filtered, washed, and centrifuged before being re-suspended.

Collectively, enzymatic dissociation methods offer easier handling and yield a higher number of intact nuclei compared to other techniques, as they involve fewer tissue fragments and eliminate the need for multi-step filtration. However, nuclei obtained through this method are primarily used for FISH and DNA copy number analysis. This limited usage for transcriptome analysis can be attributed to the considerable degradation of RNA within the nucleus caused by prolonged high-temperature treatment of FFPE samples. In addition, prolonged exposure to enzyme buffers may increase the permeability of the nuclear membrane, resulting in RNA molecule leakage and adversely affecting snRNA-seq experiments conducted in droplets. On the contrary, the mechanical homogenization strategies cause less damage to RNA molecules within the nucleus and are more suitable for snRNA-seq compared to traditional methods. However, the homogenization of formaldehyde-fixed tissue poses challenges, leading to the presence of debris in the resulting nuclei suspension, which necessitates multiple filtration steps. This, in turn, affects the yield of nuclei and may result in the loss of smaller nuclei. Combining both enzymatic dissociation and mechanical extraction methods could potentially mitigate the drawbacks associated with each technique, preserving RNA integrity while improving nuclear purity, thereby aligning more effectively with the requirements of snRNA-Seq. However, the presence of tissue fragments remains a challenge, introducing a higher amount of ribosomal RNA (rRNA) and necessitating additional steps for rRNA elimination during snRNA-seq, which can affect sequencing data quality.

## 3. Potential Development of Single-Nucleus Sequencing Technologies for FFPE Tissues

In recent years, high-throughput sc/snRNA-seq methods have revolutionized the field of biomedical research. The accurate characterization of transcriptomics in single cells within clinical FFPE specimens holds great promise for enhancing our understanding of cell heterogeneity and population dynamics, thereby improving the precision diagnostics, treatment, and prognosis of human disease. However, existing high-throughput sc/snRNA-seq platforms are not suitable for PFA-fixed and FFPE samples. Previous transcriptome studies of FFPE samples have mainly relied on low-throughput microRNA amplification assays. Recently, 10× Genomics and M20 Genomics announced sc/snRNA-seq strategies specifically designed for FFPE samples, signaling a significant advancement in efficient transcriptomic profiling of archived FFPE samples. Here, we summarize the technologies or potential technologies available for transcriptomics research in FFPE samples and divide them into three categories (Table 2).

### 3.1. Based on A-Tailing Capture

In eukaryotes, RNA with polyadenylation (Poly-A) includes mRNA and long non-coding RNA (lncRNA), accounting for only 1–5% of the total-RNA amount. Based on the principle of complementary base pairing, researchers designed oligo-dT probes to achieve the goal of capturing polyadenylated RNA from total RNA (Figure 1A).

In 2016, Levi, B.P. [30] developed fixed and recovered intact single-cell RNA (FRISCR), which combines reverse crosslinking, dT_25_ beads for RNA purity, and Smart-seq2 to profile the transcriptomes of individual fixed, stained, and sorted cells. The results demonstrate that fixation and purification introduce minimal bias and yield gene expression data similar to that of living cells. Smart-3SEQ [31] has been reported for single-cell detection of FFPE samples (Figure 1B). However, these well plate-based methods are compatible with PFA-fixed cells, but have a relatively low throughput, limiting their applicability in the search for rare phenotypes within broad cellular populations. A high-throughput scRNA-seq method called scifi-RNA-seq (‘single-cell combinatorial fluidic indexing’ RNA-seq) [33] has been developed, which combines well plate-based combinatorial indexing with the 10× platform. It has been shown to work with formaldehyde-fixed single-cells/nuclei, although it requires a separate reverse-transcription step before droplet encapsulation, which complicates the sample processing. Another method called inCITE-seq [34] (intranuclear cellular indexing of transcriptomes and epitopes) has been developed for sequencing formaldehyde-fixed single nuclei with the 10× platform. In contrast to scifi-RNA-seq, inCITE-seq performs crosslink reversal and reverse transcription inside the droplets. However, similar to scifi-RNA-seq, inCITE-seq requires laborious preprocessing of the samples. Subsequently, Savaş, T. described FD-seq [35] (Fixed Droplet RNA sequencing), a droplet-based high-throughput method for sc/snRNA-seq of PFA-fixed cells/nuclei. This technique modifies the Drop-seq process to facilitate crosslink reversal and reverse transcription of fixed cells within the droplet, similar to inCITE-seq. However, unlike FD-seq, inCITE-seq cannot use proteinase K because reverse transcription occurs inside the droplets within the 10× platform, rendering the reverse transcriptase susceptible to digestion if proteinase K were added to the lysis buffer. Unkindly, the size of cDNA obtained by FD-seq was distributed between 600 and 700 bp [42], indicating suboptimal release of long mRNA fragments from fixed samples within the droplet. VASA-seq (vast transcriptome analysis of single cells by dA-tailing) combines Smart-3SEQ and Smart-total-seq (Figure 1C) to detect the total transcriptome in single cells/nuclei (Figure 1D). This method enables the capture of both non-polyadenylated and polyadenylated transcripts across their length, using both plate and droplet microfluidic formats. However, the implementation of high-throughput detection using this technology requires the establishment of two sets of microfluidic systems. In 2022, Regev, A. et al. presented snFFPE-Seq [10], a method for snRNA-Seq (10× Genomics) of FFPE samples (Figure 1A). This method optimizes multiple stages of the process, including tissue deparaffinization and rehydration, intact nucleus extraction, and decrosslinking and deproteinization, for both plated-based and droplet-based snRNA-seq. They applied an approach for FFPE nuclei involving the utilization of thermolabile proteinase K to deproteinize and de-crosslink nucleus suspensions extracted from FFPE tissue at room temperature. Subsequently, the proteinase was heat inactivated, and the nuclei were partially reverse crosslinked before being loaded onto a droplet-based platform. However, although snFFPE-Seq is currently the simplest method to achieve snRNA-Seq of FFPE samples, it is constrained by the characteristics of A-tailing capture. As a result, this technique is limited to fresh fixed or fresh FFPE samples and lacks the sensitivity required for long-term preserved samples in archives.

### 3.2. Targeted Probe Capture

While poly-A capture presently stands as the prevailing mRNA enrichment method in commercial platforms, its utility encounters constraints in scenarios where sample degradation occurs. In contrast to poly-A capture, the gene probe method engages in molecular hybridization with the target gene, yielding a hybridization signal. This signal facilitates the visualization of the target gene within the expansive genome, thus emerging as a potent approach for investigating transcriptomics in samples characterized by extensively fragmented RNA. The gene probe capture technology is designed to apprehend and hybridize target mRNA fragments within the nucleus, rendering it particularly suitable for samples comprised of degradable tissues.

TempO-Seq (Templated Oligo assay with Sequencing readout) [37], a ligation-based targeted whole-transcriptome expression profiling assay, was used to identify previously unreported compound-responsive genes and incorporate them into a comprehensive, yet specific, compound signature (Figure 2A). MLPA-seq [43] is an enhanced version of the well-known MLPA (Multiplex Ligation-dependent Probe Amplification) method. Both approaches overcome the limitations of the original MLPA in terms of multiplexing and detection. They leverage the sensitivity of NGS (next-generation sequencing) to analyze up to 20,000 target RNA. However, these methods overestimate the original number of studied molecules and suffer from bias caused by PCR amplification. In contrast, the TAC-seq (Targeted allele counting by sequencing) [39] method employs UMIs to estimate the original molecule counts of mRNAs and can overcome amplification bias in NGS and maximize nucleic acid detection sensitivity (Figure 2B). Nonetheless, none of the above techniques can facilitate high-throughput snRNA-seq of FFPE samples. Recently, 10× Genomics introduced Chromium Fixed-RNA profiling (Fixed-RNA) for FFPE samples (Figure 2C). This approach enables gene expression profiling of thousands to hundreds of thousands of cells or nuclei using a sensitive probe-based method. It captures the entire transcriptome of humans or mice, including genes with low expressing levels. The Fixed-RNA protocol uses two pairs of specific probes designed for the target mRNA, covering 18,000 genes in humans and 20,000 genes in mice. In detail, the fixed samples (cells or nuclei) were permeabilized, followed by the hybridization of barcoded probes to the mRNA within the cell/nucleus. Once the probe is hybridized, only two consecutive probes of the target mRNA can hybridize simultaneously, resulting in a probe connection. After the mRNA template is completely degraded, the paired probes are released from the nucleus. These probes are then captured by Gel Beads, which contain cell barcode, UMI, and pCS1 sequences. Finally, the probe sequence carrying the barcode is used for library construction and sequencing. Moreover, the Fixed-RNA technology is optimized with a microfluidic chip, allowing for the reuse of up to 16 samples per channel. This optimization enables increased throughput and expands the capacity to profile large numbers of samples, while reducing the cost per sample. The technique called snPATH-seq (single nuclear pathology sequencing) [9], which emerged around the same time as snFFPE-Seq, applies Fixed-RNA to the nuclear transcriptome of FFPE samples. Furthermore, Taylor et al. [11] analyzed FFPE human breast cancer sections using Fixed-RNA and spatial transcriptomics (10× Visium). The integration of these technologies allows for deeper insights, facilitating discoveries that advance oncology research and the development of diagnostics and therapeutics. However, limited by the types of gene probes, this technology can only capture known genes. Moreover, the technique cannot be applied to species other than humans and mice at present.

### 3.3. Random Primer Capture Technique

The random primer method is a common molecular biotechnology technique that is mainly used to amplify specific RNA fragments. Due to the arbitrariness of random primers, RNA sequences of any length can be amplified and, in turn, be suitable for transcriptomic studies of FFPE samples.

Split Pool Ligation-based Transcriptome sequencing (SPLiT-seq) [40], a scRNA-seq method that labels the cellular origin of RNA through combinatorial barcoding, has been successfully used in fixed cells using random primers that were more efficient and broader to capture total RNAs [44]. scFAST-seq (single-cell Full-length RNA Sequence Transcriptome sequencing) [41], a method that combines semi-random primers with high reverse-transcription efficiency, template switching, and convenient rRNA removal methods, allows the construction of full-length RNA libraries of up to 12,000 cells within 8 h (Figure 3A). Although this technique is not currently compatible with FFPE samples, it has the potential to be applied to fixed or paraffin-embedded samples. In parallel with Fixed-RNA (10× Genomics), M20 Genomics has launched a single-nucleus full-length transcriptome amplification technology based on random primers, which can amplify transcripts from RNA incomplete samples and is suitable for snRNA-seq in FFPE samples. Based on this approach, Wang and Guo et al. developed a droplet-based snRNA sequencing technology (snRandom-seq) [12] for FFPE tissues by capturing full-length total RNAs with random primers, which provides a powerful snRNA-seq platform for clinical FFPE specimens (Figure 3B). Briefly, in a single nucleus, the total RNA was captured for reverse transcription using random primers, and the second strand was synthesized by performing poly-dA tailing on the first-strand cDNAs. cDNAs were then specifically tagged using a microfluidic barcoding platform, followed by amplification and sequencing. This innovative technique combines VASA-seq and scFAST-seq to ultimately achieve oligo-dT capture of total RNA. However, the technique is currently more suitable for the nucleus because of the larger proportion of rRNA in cells.

## 4. Current Limitations and Future Developments

FFPE tissues represent a substantial and valuable repository of patient materials encompassing essential clinical history and follow-up data. Although researchers have gradually become aware of the potential for obtaining expression profiles of individual cells or nuclei FFPE tissues, this approach remains challenging. 

The first challenge arises from the extraction or isolation of high-quality nuclei. The isolation and application of FFPE nuclei has a long history, and spatial transcriptome (ST) technology has found extensive application in FFPE samples. However, little analysis of the nuclear transcriptome has been reported, potentially indicating the inapplicability of FFPE nuclei in transcriptomic profiling. Tissue homogenization, even in fixed brain tissue, becomes increasingly challenging due to molecular crosslinking within the nuclei. In addition, effectively removing excessive tissue debris poses difficulties and leads to severe contamination of the snRNA-seq data. Moreover, the high proportion of rRNA requires additional removal processes when employing total-RNA protocols. Currently, most extraction methods employed in high-throughput snRNA-seq platforms rely on a combination of enzyme dissociation and homogenate. Despite the optimization of nuclear suspension and RNA quality within the nucleus, significant limitations still persist. One notable drawback is the requirement for a wider range of reagents and increased filtration when combining the two methods, which ultimately diminishes the cell nucleus yield. In addition, the enzymatic treatment at high temperatures leads to RNA molecule leakage, thereby affecting the detected transcripts. Furthermore, all current methods for preparing nuclei from FFPE samples focus on tissue sections (5–100 μm), while disease research often involves the tissue blocks. Mechanical methods prove inadequate for the preparation of large tissue nuclei, as they introduce more tissue debris. In contrast, enzymolysis methods yield purer nuclei; thus, it is crucial to explore a method that can obtain a substantial quantity of pure nuclei, while minimizing RNA molecule leakage.

Another challenge lies in the utilization of microfluidic technology platforms for snRNA-seq, whereby only a subset of cells in the suspension are captured as they are automatically and randomly separated into droplets or wells. This approach creates an illusion that millions of nuclei can be dissociated from the tissues, but only approximately 10,000 are trapped. Consequently, it raises concerns about the ability to accurately reflect the true cell composition. One potential solution to this issue is to reduce the speed at which cells are captured. Furthermore, considerable variations exist in the results (including the number of nuclei, genes, UMIs, and others) obtained from the same commercial platform, even when the same experimenter processes samples from the same batch. These discrepancies are particularly pronounced when comparing results across different companies. The major contributing factor to these inconsistencies is the inherent randomness and low capture rate associated with nucleus isolation. Although it is possible to calibrate the data by conducting batch-based adjustments, our previous experiments have shown that data obtained from the same samples and same platforms, but using two different companies, sometimes cannot be combined for analysis.

For fresh FFPE samples, poly-A capture of mRNA can fulfill the needs of researchers, especially for non-tumor samples. This method offers two technical approaches: (1) heating the nucleus within the droplet, followed by reverse crosslinking and reverse transcription in the same droplet; and (2) reverse crosslinking the nuclei externally to the chip prior to encapsulation in droplets. Currently, the first approach requires the inclusion of proteinase K in the lysate, which limits the reverse-transcription reaction. Exploring a reverse-crosslinking lysate devoid of this enzyme is essential to enable direct compatibility of FFPE sample nuclei with conventional 3′ sequencing platforms, ensuring ease of operation and cost-effectiveness. However, this is not sensitive enough to capture the low-quality RNAs from FFPE tissues. The second approach proves more suitable for FFPE samples, as it can be compatible with a wider range of platforms, such as scFAST-seq and VASA-seq, enabling comprehensive transcript detection of total RNA. Considering the degradation characteristics of RNA molecules in FFPE samples, full-length transcriptome sequencing technologies, like Fixed-seq and snRandom-seq, offer greater suitability. However, existing reports predominantly employ FFPE samples stored for up to six months, necessitating further exploration of the applicability of these techniques to long-term stored samples. 

Moreover, the absence of standardized guidelines for processing FFPE tissue samples from patients to preserve biomolecules has posed a significant challenge. While FFPE samples have proven effective for histology, the lack of standardized guidelines has hindered their utilization in molecular analyses. There exists considerable variation in tissue fixation times and methods among hospitals, laboratories, and service companies [5]. Our previous study showed that the fixed duration of brain tissues has pronounced negative effects on the transcriptional profiles of its nucleus, and a cliff-like effect appears within 1 to 3 days of the fixation time [45]. Various factors can influence the quality of nuclear RNA in FFPE tissues, including postmortem time, fixation distance, sample storage conditions (e.g., temperature), dehydration, hydration, paraffin incubation time, and temperature for different tissue sizes, all of which impact RNA integrity for transcriptomic research. Previous studies have shown that large tissue specimens may temporarily sustain cellular viability due to the slow penetration of PFA, resulting in cellular autolysis [46]. Therefore, it is critical to explore a standardized process to maximize RNA protection and minimize degradation during FFPE sample preparation.

Single-cell/nucleus and spatial transcriptome are revolutionizing the resolution of molecular states in clinical tissue samples, offering a comprehensive understanding of cancer biology by unraveling the complexities of the tumor microenvironment [11,47]. Currently, the 10× Genomics Chromium (single cell/nucleus) and Visium platforms complement each other: Chromium data lack spatial context, while Visium data require integration with single-cell/nucleus data to infer detailed information about cell type composition. The combination of snRNA-seq and ST technologies in FFPE samples is expected to become standard practice, with the emergence of spatial total RNA-seq [48] facilitating the exploration of spatial information within FFPE samples. Moreover, the future development of multi-omics techniques specific to FFPE tissues holds promise.

## 5. Conclusions

In summary, snRNA-seq opens the era of great navigation for FFPE tissue. In this perspective, we have summarized the nuclear preparation methods for FFPE samples over the recent decades, and discussed the limitations and prospects of these methods for transcriptomics analysis of FFPE samples. Furthermore, we provided an overview of snRNA-seq techniques specifically designed for FFPE samples, examining their advantages and limitations. While the current published papers mainly focus on two commercial platforms, it is important to note that the available snRNA-seq techniques suitable for FFPE samples extend beyond these platforms. Therefore, we explore the potential technical prospects in this review. With the advent of a new era of high-throughput snRNA-seq for FFPE tissues, researchers are progressing towards a more complete understanding of tissue transcriptome properties within human tissue archives.

## Figures and Tables

**Figure 1 ijms-24-13744-f001:**
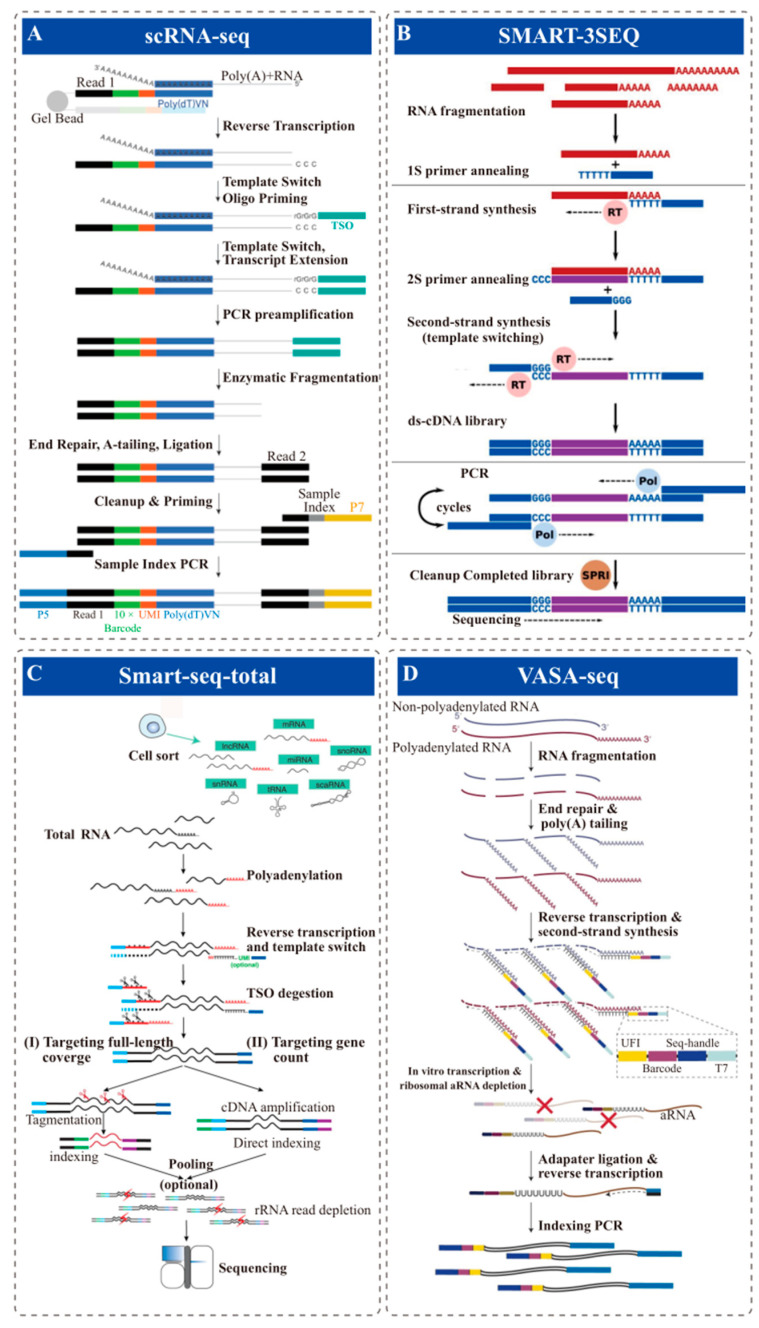
sc/snRNA-seq of FFPE samples based on Oligo-dT probe capture techniques. (**A**) Traditional 3′ sequencing library construction principle (ex., 10×, drop-seq, AccuraCell™). (**B**) Conceptual diagram of the Smart-3SEQ library preparation method. Reprinted with permission from ref. [31]. Copyright 2019 Genome Research. RNA is denatured and fragmented by hydrolysis; The oligo(dT) primer captures poly-dA; Synthesizes first-strand cDNA and the second cDNA strand; PCR with long primers amplifies and cleanup step uses SPRI beads; The final library contains the unknown cDNA sequence between the two sequencing adapters. (**C**) Schematic of Smart-seq-total pipelines. Reprinted with permission from ref. [32]. Copyright 2021 PNAS. Cell lysis, total RNA is polyadenylated, primed with anchored oligo-dT; After reverse transcription, TSO is enzymatically cleaved, single-stranded cDNA is amplified and cleaned up; Amplified cDNA is then either tagmented or directly indexed, pooled, and depleted from ribosomal sequences. (**D**) Overview of the VASA-seq single-cell molecular workflow. Reprinted with permission from ref. [36]. Copyright 2022 Nature biotechnology. Single cells are lysed, and RNA is fragmented; Fragments are repaired and polyadenylated, followed by reverse transcription (RT) using barcoded oligo-dT primers; The cDNA is made double stranded and amplified using IVT; aRNA is depleted of rRNA, and libraries are finalized by ligation, RT and PCR, which leave fragments ready for sequencing.

**Figure 2 ijms-24-13744-f002:**
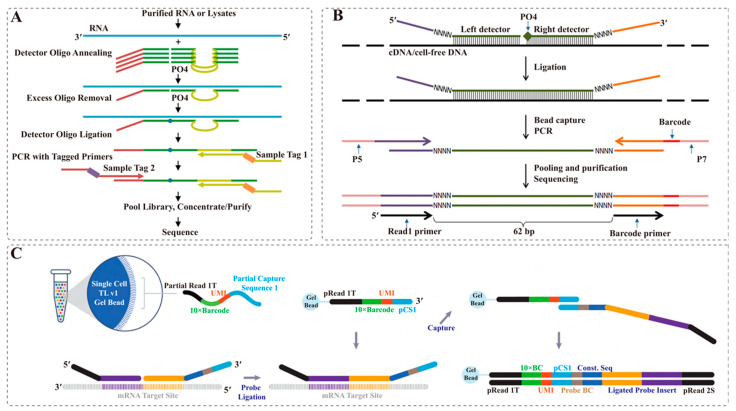
sc/snRNA-seq of FFPE samples based on gene probe capture techniques. (**A**) TempO-Seq biochemical LOSscheme. Reprinted with permission from ref. [37]. Copyright 2017 PLOS ONE. RNAs are targeted by annealing to DOs that contain target-specific sequences (green), as well as primer landing sites (red and yellow) that are shared across all Dos; Excess oligos are removed by a 30 exonuclease, then the hybridized oligos are ligated and amplified using primers that contain sample tag (index) sequences (orange and purple bars) and adaptors required for sequencing. (**B**) Schematic diagram of the assay to detect specific mRNA or cell-free DNA. Reprinted with permission from ref. [39]. Copyright 2018 Genomic Medicine. Target-specific DNA oligonucleotide detector probes hybridize under stringent conditions to the studied cDNA or cfDNA; Both detector oligonucleotides consist of a specific 27 bp region (green), 4 bp unique molecular identifier (UMI), and universal sequences (purple and orange); The right detector oligonucleotide is 5′ phosphorylated. After rigorous hybridization, the pair of detector probes is ligated using a thermostable ligase under stringent conditions; Next, the ligated detectors complexed with the target region are captured with magnetic beads and PCR amplified to introduce sample-specific barcodes and other common motifs that are required for single-read NGS. (**C**) Chromium Single-Cell Fixed-RNA Profiling library construction principle. Cells are fixed and permeabilized; Samples are hybridized to probe sets and may be processed individually or pooled with up to 16 samples in a single lane of a Chromium chip; During GEM generation, the probe sets are ligated and extended to incorporate unique barcodes. Sequencing libraries are then prepared, sequenced, and analyzed.

**Figure 3 ijms-24-13744-f003:**
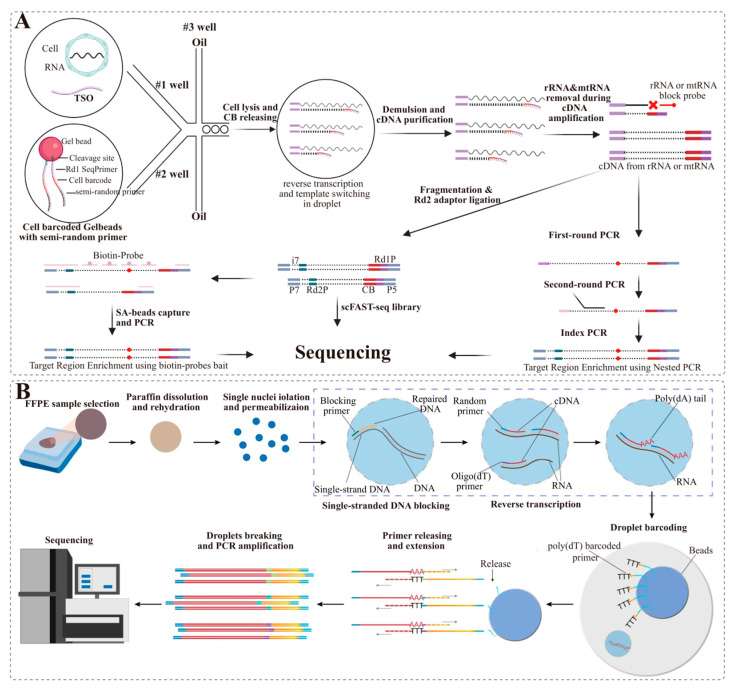
sc/snRNA-seq of FFPE samples based on random probe capture techniques. (**A**) Library construction principle of scFAST-seq. Cells were suspended in reverse-transcription master mix and then encapsulated into droplets with cell-barcoded gel beads; RNA released from each cell was reverse transcribed to cDNA and barcoded by the corresponding semi-randomers and tailed by template switching oligos within droplets; After breaking the droplets, pooled cDNA from all droplets was purified and amplified, while amplification of cDNA derived from rRNA and mtRNA was inhibited by blocking probes; Then, constructing the library. (**B**) The workflow of snRandom-seq for FFPE tissues. Reprinted with permission from ref. [12]. Copyright 2023 Nature Communications. FFPE sample selection, paraffin dissolution, single-nuclei isolation, and permeabilization, single-strand DNAs blocking, reverse transcription, dA tailing, droplet barcoding, primers releasing and extension, droplets breaking and PCR amplification, and sequencing.

**Table 2 ijms-24-13744-t002:** Detailed comparison of current and potential methods and technologies of sc/snRNA-seq for FFPE samples.

Approach	Technology	Input	Samples	Throughput	Target	SN or SC?	Genes Measured	Reads/Sample	UMI	Full-Length	Refs
Based on dA-tailing	Pick-Seq	FFPE	Human tonsil, breast cancer	Low	Poly-A RNA	5~10 cells	2700, 4640	/	No	No	[29]
	FRISCR	Fixed	hESCs	Low	Poly-A RNA	SC	12,000	/	No	Yes	[30]
	Smart-3SEQ	FFPE	DCIS	Low	Poly-A RNA	No	~3000	/	No	No	[31]
	Smart-total-seq	Cell lines	Fibroblasts, HEK293T, and MCF7	Low	Total RNA	SC	/	/	No	Yes	[32]
	scifi-RNA-seq	Fixed	Clone E6-1	High	Poly-A RNA	SN	/	/	2000~6000	No	[33]
	inCITE-seq	Fixed	Hela	High	Poly-A RNA	SN	1158	/	2655	No	[34]
	FD-seq	Fixed	A549 cells	High	Poly-A RNA	SC	~3000	58.6 K	~8000	No	[35]
	VASA-seq	Frozen	HEK293T and mESCs	High	Total RNA	SC	~10,000	750 K	/	Yes	[36]
	snFFPE-Seq	FFPE	Brain	High	Poly-A RNA	SN	276	154	/	No	[10]
Based on targeted probe	TempO-Seq	Cell lines	Reference RNA	Low	Target mRNA	No	2244	1.9 M	No	No	[37,38]
	TAC-seq	FF	Endometrial biopsies	Low	Target mRNA	No	/	/	Yes	No	[39]
	Fixed-RNA	FFPE	Breast	High	Target mRNA	SN	1850	/	/	No	[9,11]
Based on random primer	SPLiT-seq	Fixed	293T, 3T3, and Hela-S3	High	Total RNA	SN	~5000	>500 K	12,000~15,000	No	[40]
	scFAST-seq	FF	K562, A549, and HCC827	High	Total RNA	SC	~1000	/	~2000	Yes	[41]
	snRandom-seq	FFPE	293T, 3T3, and some organs	High	Total RNA	SN	~4000	25~29 K	~10,000	Yes	[12]

Total RNA: RNA other than rRNA; Fixed: fixation with formaldehyde or paraformaldehyde; FF: fresh frozen; FFPE: formalin-fixed paraffin-embedded; SC: single cell; SN: single nuclei; hESCs: human embryonic stem cells; DCIS: ductal carcinoma in situ; HEK293T: human embryonic kidney 293; MCF7: breast cancer cells; Clone E6-1: human Jurkat cells; A549: human lung adenocarcinoma cells A549; mESCs: mouse embryonic stem cells; K562: chronic myeloid leukemia cancer cells; HCC827: human non-small-cell lung cancer cells.

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
