# Peer review of "Single-Nucleus RNA-Seq: Open the Era of Great Navigation for FFPE Tissue"

_ijms, 2023, doi:10.3390/ijms241813744_

Round 1

Reviewer 1 Report

The manuscript is a critical review of an emerging useful technique of sn-RNA-seq of nuclei isolated from formalin fixed paraffin-embedded tissue blocks. The ability to derive informative data from such samples, that are readily available in huge numbers  may be of great value. The present review is timely and  maybe have be of useful source of information for researchers that intend  to use sn-RNA-seq for their studies. The readers can find information on how to prepare nuclei from FFPE scrolls  and the pros and cons of the different single nucleus RNA purification and sequencing technologies.  

However, before publication the authors should pay attention to the following points.

Major points

The authors should pay more attention to the figures. All three figures need to be of better quality (better resolution). As they are presented are too small to be useful, especially the fonts. In most cases the figures are copies from published figures and may be that is the problem in reproduction. In some cases the copy rights are mentioned but in other they are not.

Minor points

In the introduction, first page, second paragraph, the last sentence is too long which makes it difficult to follow and even more the ‘,which is only available weeks to years after…’ makes hard to understand what the author want to convey.

Page 3 first sentence ‘FFPE nuclear’ most likely the authors mean ‘FFPE nuclei’.

Page 3 line 6 ‘CaCL2’ need to be ‘CaCL2’.

Page 4 Starting from section 3 until page 9 the font size is increased. The same in page 12 section5 Conclusion.

Author Response

Point 1: The authors should pay more attention to the figures. All three figures need to be of better quality (better resolution). As they are presented are too small to be useful, especially the fonts. In most cases the figures are copies from published figures and may be that is the problem in reproduction. In some cases the copy rights are mentioned but in other they are not.

Response 1: Thank you very much for your suggestion. We have modified the font size of all figures in the manuscript, especially Figure1. Some figures in the current manuscript derived from published articles reproduction were downloaded with a copyright license and sent to the editor at the time of initial submission. In addition, the figures of the copied articles are indicated as sources in the notes section, and the rest without the source are reprocessed and drawn by ourselves.

Point 2: In the introduction, first page, second paragraph, the last sentence is too long which makes it difficult to follow and even more the ‘, which is only available weeks to years after…’ makes hard to understand what the author want to convey.

Response 2: Thank you for your suggestion. We have polished the manuscript as shown below.

Revision: Line 40-46: “Formalin-fixed paraffin-embedded (FFPE) tissue blocks represent the paramount approach for preserving human tissue in clinical diagnostics. Globally, pathology laboratories and sample banks house over a billion FFPE sections. These repositories offer a precious resource for profound transcriptomic analysis, albeit accessible only weeks to year post-sample collection. This delay is necessitated by critical clinical attributes such as tumor genetics, treatment response, and patient survival, which require adequate time for meaningful development.”

Point 3: Page 3 first sentence ‘FFPE nuclear’ most likely the authors mean ‘FFPE nuclei’.

Response 3: The first sentence 'FFPE nuclear' on Page 3 should indeed be amended to 'FFPE nuclei', which we have corrected it.

Revision: Line 84: “Various methods exist for the isolation of FFPE nuclei, with the majority being employed for FISH.”

Point 4: Page 3 line 6 ‘CaCl2’ need to be ‘CaCl2’.

Response 4: According to your suggestion, we have corrected the ‘CaCl2’ to ‘CaCl2’.

Revision: Line 88: “Specifically, FFPE scrolls were de-crosslinked by heating in an EDTA solution, followed by washing with CaCl2, then overnight digestion in a mixed dissociation solution of multiple enzymes (collagenase type 3, purified collagenase, and hyaluronidase) was performed.”

Point 5: Page 4 Starting from section 3 until page 9 the font size is increased. The same in page 12 section5 Conclusion.

Response 5: Thanks for your suggestion, we have revised the font size in the manuscript uniformly.

Reviewer 2 Report

In this review, Guo et al., present an overview of single-nucleus RNA sequencing and the application of this technology to formalin-fixed paraffin-embedded. The manuscript is well written and the authors provide all the steps and the protocols for isolation, digestion and FFPE tissues manipulation including distinct approaches (pick-Seq, FD-seq, TAC-seq etc). Some improvements should be included: 

It is difficult for the reader to understand Figure 1. Please increase the quality i.e A and B in one row and C and D below, increase the letters, etc..

As this is a review, the authors could briefly explain in the initial paragraph of 3.2 the difference between A-tailing capture and targeted probe capture before explaining the second one.

Comparing the two previous approached as well as random probe capture, the authors could highlight the quality control metrics across the protocols due to differences in tissue type recovery and in sequencing depth between preparations?

The authors could also stress details/workflows/analytical pipelines relatively to the computational protocol(s) of the provided data based on snRNA-seq and comparisons of gene expression signatures using scRNA-seq and snRNA-seq.

Minor editing of English language required

Author Response

Point 1: It is difficult for the reader to understand Figure 1. Please increase the quality i.e A and B in one row and C and D below, increase the letters, etc..

Response 1: Thank you for your suggestion, we have rearranged the graph in Figure 1 and changed the font size.

Point 2: As this is a review, the authors could briefly explain in the initial paragraph of 3.2 the difference between A-tailing capture and targeted probe capture before explaining the second one.

Response 2: According to your suggestion, we briefly explain the difference between A-tailing capture and targeted probe capture in the initial paragraph of 3.2.

Revision:

Line 199-205: “While poly-A capture presently stands as the prevailing mRNA enrichment method in commercial platforms, its utility encounters constraints in scenarios where sample degradation occurs. In contrast to poly-A capture, the gene probe method engages in molecular hybridization with the target gene, yielding a hybridization signal. This signal facilitates the visualization of the target gene within the expansive genome, thus emerging as a potent approach for investigating transcriptomics in samples characterized by extensively fragmented RNA. The gene probe capture technology is designed to apprehend and hybridize target mRNA fragments within the nucleus, rendering it particularly suitable for samples comprised of degradable tissues.”

Point 3: Comparing the two previous approached as well as random probe capture, the authors could highlight the quality control metrics across the protocols due to differences in tissue type recovery and in sequencing depth between preparations?

Response 3: Thanks for your constructive comments, we have added tables related to different technologies and quality control metrics in the manuscript (Table 2).

Point 4: The authors could also stress details/workflows/analytical pipelines relatively to the computational protocol(s) of the provided data based on snRNA-seq and comparisons of gene expression signatures using scRNA-seq and snRNA-seq.

Response 4: Thank you very much for your suggestion. Due to the focus of this review on FFPE sample nuclear separation strategies and their existing and potential snRNA-seq techniques, there is no concentrated space to describe the workflow and bioinformatics analysis pipeline of single nuclear sequencing. Numerous bioinformatics tools employed for scRNA-seq analysis can be readily adapted for snRNA-seq analysis, encompassing quality control, clustering, cell type annotation and other essential processing steps (cell communication, pseudotime, RNA velocity, etc). The current snRNA-seq used for FFPE samples can capture full-length transcripts or total RNAs, which is helpful for RNA velocity analysis. Moreover, these full-length total snRNA-seq datasets allow a comprehensive analysis of copy number variation, alternative splicing, and mutations at single-cell/ nucleus level. As far as I know, the current software can still analyze such data. With the diversity of FFPE technologies, a large amount of more complex and precise data will emerge, and existing analysis methods may not be able to meet the requirements, especially for full-length and full-transcript data.

Reviewer 3 Report

"Single-nucleus RNA-seq: open the era of great navigation for FFPE tissue".

This paper discusses the significance of single-cell sequencing techniques, particularly single-nucleus RNA-sequencing (snRNA-seq), in the context of exploring heterogeneity and genetic variations at the single-cell level. The potential of snRNA-seq to enable the exploration of many FFPE tissues is highlighted. The review describes the strategies for nuclear preparation of FFPE samples and the latest technologies of snRNA-seq related to FFPE specimens. The authors emphasize the feasibility of these techniques for future studies utilizing FFPE samples in transcriptomics analysis.

The review then discusses the limitations of snRNA-seq in FFPE samples and outlines potential avenues for technical developments in this area. It emphasizes that snRNA-seq holds the potential to revolutionize the analysis of FFPE tissues and offers a comprehensive understanding of tissue transcriptome properties within human tissue archives. It acknowledges that while current research mainly focuses on specific commercial platforms, the potential of snRNA-seq extends beyond these platforms, thus hinting at a broader horizon for technical innovations.

I find this review to be both timely and of significant importance, making it suitable for publication in the International Journal of Molecular Sciences (IJMS) after minor revisions.

1) One key consideration pertains to the fundamental relationship between scRNA-seq and snRNA-seq (not concerning FFPE). While scRNA-seq primarily assesses cell-level transcript abundance, snRNA-seq is more closely associated with direct determination of gene expression per se. This distinction arises from snRNA-seq's ability to capture 'freshly' transcribed RNA within nuclei. A valuable discussion on this aspect could be included.

2) The second aspect involves the technical interplay of 'fresh' scRNA-seq and FFPE snRNA-seq, encompassing their advantages and disadvantages. In the Abstract, the authors introduced FFPE snRNA-seq as a promising alternative to scRNA-seq due to its reduced ionized transcription bias and compatibility with more diverse samples. Furthermore, in the first paragraph of the Introduction, the authors highlighted the drawbacks of the 'fresh' method and the benefits of the FFPE method.

However, in the second paragraph of the Introduction, they said, "Although fresh or fresh frozen clinical samples are ideal for transcriptomic analysis, the limited availability of these samples is a serious drawback." This implies that scRNA-seq using fresh (frozen) samples is considered the gold standard. Furthermore, the "4. Current limitations and future developments" section outlined numerous shortcomings of the FFPE method.

Therefore, a crucial query arises: is the FFPE method inherently superior to the fresh (frozen) method, or is its primary advantage solely linked to its applicability to a huge archive of FFPE samples? A more explicit response to this question is needed. If the advantage of FFPE method solely pertains to archive of sample, the authors should temper their enthusiastic portrayal of the FFPE method in the Abstract and Introduction, avoiding an undue implication that it is superior to the fresh (frozen) method.

 3) It would be pertinent to mention a flow cytometry approach (PubMed: 7743898), which distinguishes cells, which were viable or dead at the moment of fixation by formalin, based on their nuclear states at the moment of fixation. By employing flow cytometry sorting to isolate nuclei from viable cells, the noise caused by nuclei from dead cells can be removed.

 4) The inconsistency in font sizes of different paragraphs is distracting.

English is quite good.

Author Response

Point 1:  One key consideration pertains to the fundamental relationship between scRNA-seq and snRNA-seq (not concerning FFPE). While scRNA-seq primarily assesses cell-level transcript abundance, snRNA-seq is more closely associated with direct determination of gene expression per se. This distinction arises from snRNA-seq's ability to capture 'freshly' transcribed RNA within nuclei. A valuable discussion on this aspect could be included.

Response 1: Thank you for your constructive comments. As you said, snRNA-seq's ability to capture 'freshly' transcribed RNA within nuclei, snRNA-seq is more closely associated with direct determination of gene expression per se, so in some ways, snRNA-seq can be substituted for scRNA-seq. We described it in the first paragraph and last sentence of the introduction, and your suggestion in another way further supports the importance of snRNA-seq.

Revision 1: Line39-42: “In addition, owing to its ability to capture newly transcribed transcripts within the cell nucleus, snRNA-seq establishes a more direct correlation with gene expression. Consequently, snRNA-seq demonstrates superior applicability over scRNA-seq in specific contexts.

Point 2:  The second aspect involves the technical interplay of 'fresh' scRNA-seq and FFPE snRNA-seq, encompassing their advantages and disadvantages. In the Abstract, the authors introduced FFPE snRNA-seq as a promising alternative to scRNA-seq due to its reduced ionized transcription bias and compatibility with more diverse samples. Furthermore, in the first paragraph of the Introduction, the authors highlighted the drawbacks of the 'fresh' method and the benefits of the FFPE method.

However, in the second paragraph of the Introduction, they said, "Although fresh or fresh frozen clinical samples are ideal for transcriptomic analysis, the limited availability of these samples is a serious drawback." This implies that scRNA-seq using fresh (frozen) samples is considered the gold standard. Furthermore, the "4. Current limitations and future developments" section outlined numerous shortcomings of the FFPE method.

Therefore, a crucial query arises: is the FFPE method inherently superior to the fresh (frozen) method, or is its primary advantage solely linked to its applicability to a huge archive of FFPE samples? A more explicit response to this question is needed. If the advantage of FFPE method solely pertains to archive of sample, the authors should temper their enthusiastic portrayal of the FFPE method in the Abstract and Introduction, avoiding an undue implication that it is superior to the fresh (frozen) method.

Response 2: Thank you for your comments.  In the abstract, we want to express that snRNA-seq can be used as an alternative technique for the study of biological tissue transcriptomics, regardless of sample type: “Line 11-15: Single-cell sequencing (scRNA-seq) has revolutionized our ability to explore heterogeneity and genetic variations at the single-cell level, opening up new avenues for understanding disease mechanisms and cell-cell interactions. Single-nucleus RNA-sequencing (snRNA-seq) is emerging as a promising solution to scRNA-seq to its reduced ionized transcription bias and compatibility with richer samples.” Based on the advantages of snRNA-seq, we hypothesize that this technique lays a foundation for exploring the transcriptomics of FFPE samples: “Line15-16: This approach will provide an exciting opportunity for in-depth exploration of billions of formalin-fixed paraffin-embedded (FFPE) tissues.”

Since FFPE samples are the primary sample type for clinical tissue preservation, rather than cryopreserved samples, Therefore, the clinical application of frozen/fresh samples in transcriptomics is limited. We will revise the first sentence of the second paragraph of the introduction to make it more appropriate.

Currently, studies applicable to the transcriptomics of FFPE samples are in their infancy, and in order to verify the applicability of the technique, it is necessary to use the scRNA-seq data of fresh (frozen) samples as the gold standard. Because of the preliminary development of this technology, we discussed more about the shortcomings of current technology applied to FFPE samples and potential new technologies in the section of "4. Current limitations and future developments". These shortcomings are technical deficiencies, not deficiencies in the application of snRNA-seq in FFPE samples.

Finally, thank you very much again for your advice. At present, the technology developed by the two commercial platforms for FFPE sample snRNA-seq is indeed superior to the traditional 3 'database sequencing in terms of technical principle. For example, the snRandom-seq based on M20 company can detect the full length and full transcriptome of multiple types of samples. In addition, 10x Genomics' Fixed-RNA technoloyl captures target mrnas. What these two techniques have in common is that they can be applied to degraded samples without the unavailability caused by severe 3 'fragmentation caused by sample degradation. However, if the sample type is fresh/frozen tissue, the traditional techniques can also be used, because the new technique requires a larger nucleus input and the procedure is more complex, requiring the cells/nuclei to be fixed before the experimental process can continue.

Based on this review of the methods and techniques of nuclear lysis of FFPE samples, it does not imply that FFPE is superior to frozen/fresh samples. According to your suggestions, we will tone down this suggestive language in the text.

Revision 2: Line 43-49: “Single cell/nuclei transcriptional profiling reveals cell heterogeneity and clinically relevant traits in intra-operatively collected patient-derived tissue, but is currently limited to either freshly harvested human tissues or fresh-frozen samples.”

Point 3:   It would be pertinent to mention a flow cytometry approach (PubMed: 7743898), which distinguishes cells, which were viable or dead at the moment of fixation by formalin, based on their nuclear states at the moment of fixation. By employing flow cytometry sorting to isolate nuclei from viable cells, the noise caused by nuclei from dead cells can be removed.

Response 3:  Thanks for your suggestion, we have cited this conclusion in the section of nuclear isolation strategies.

Revision 1: Line 91: “Refenence 18”

Point 4:  The inconsistency in font sizes of different paragraphs is distracting.

Response 4: Thanks for your suggestion, we have revised the font size in the manuscript uniformly

Reviewer 4 Report

In a manuscript by Guo et al., the Authors have summarized the nuclear preparation methods for FFPE samples over the recent decades, and discussed the limitations and prospects of these methods for transcriptomics analysis of FFPE samples. The review manuscript is well written and presented. The selection of the literature is appropriate.

I have only minor editing comments:

1. page 3 - should be: CaCl2

2. Table 1 (title) - enzymatic -> Enzymatic (please apply for other Tables).

3. Figure 1 is hardly legible.

4. The manuscript is marked as "Article", however, in my opinion, this should be designated as "Review". In addition, the Authors refer to this manuscript as to the review.

Author Response

Point 1: page 3 - should be: CaCl2.

Response 1: Thank you very much for your suggestion, and we have corrected the ‘CaCl2’ to ‘CaCl2’ in page 3.

Revision: Line 89: “Specifically, FFPE scrolls were de-crosslinked by heating in an EDTA solution, followed by washing with CaCl2, then overnight digestion in a mixed dissociation solution of multiple enzymes (collagenase type 3, purified collagenase, and hyaluronidase) was performed.”

Point 2:  Table 1 (title) - enzymatic -> Enzymatic (please apply for other Tables).

Response 2: According to your suggestion, we have corrected ‘enzymatic’ in Table 1 (title) to ‘Enzymatic’, and corrected ‘summary’ in Table 2 (title) to ‘Summary’.

Revision: Line 96: “Table 1. Enzymatic dissociation strategies and applications of nuclei from FFPE tissues.” Line 141: “Table 2. Summary the potential techniques of sc/snRNA-seq for FFPE samples.”

Point 3: Figure 1 is hardly legible.

Response 3: Thank you very much for your suggestion, we have rearranged the graph in Figure 1 and changed the font size.

Point 4: The manuscript is marked as "Article", however, in my opinion, this should be designated as "Review". In addition, the Authors refer to this manuscript as to the review.

Response 4: Thank you very much for your suggestion, we have marked the manuscript as "Review".

Reviewer 5 Report

This is an interesting review. However, some points should be addressed.

-In Abstract, the autho should report the methodology and the scientific databases used to perform this revie.

- In the Abstract, the main findings of the review should be reported.

- In the Abstract, the author should add a sentences about the potential future research that should be perform on this topic.

- A Methods section about the used methodology  to collect the reviewing data should be added.

- A Flow chart diagram about the initial collected studies and the studies that finaly included in the rev should be added.

- What exclusion and inclusion criteria are the authors used to perform our review?

Minor English editing is required

Author Response

Point 1: In Abstract, the author should report the methodology and the scientific databases used to perform this revie.

Response 1: Thank you very much for your suggestion. In this review, investigation and crossbar association were used to complete the literature investigation and writing, and "Web of Science", "PubMeb" and "biorxiv" databases were used. As this review is a ‘Traditional Review’ rather than ‘Systematic Review’ or ‘Systematic Review and Meta-analysis’, this content cannot appear in the abstract of this kind of review. Thank you again for your advice.

Point 2: In the Abstract, the main findings of the review should be reported.

Response 2: Thank you for your advice. The abstract of a traditional review usually does not include the main findings, which are presented in the text content.

Point 3: In the Abstract, the author should add a sentence about the potential future research that should be perform on this topic.

Response 3: According to your suggestion, we have added the sentences about the potential future research in the abstract.

Revision: Line 21-23: The development of snRNA-seq technologies for FFPE samples will lay a foundation for transcriptomic studies of valuable samples in clinical medicine and human sample banks.

Point 4: A Methods section about the used methodology to collect the reviewing data should be added.

Response 4: Thank you for your advice. The reviewing data collected in our review came from literature from some databases. We investigated the literature on keywords such as FFPE, formaldehyde or paraformaldehyde fixed, nuclei, single-cell RNA-seq (scRNA-seq) or single-nucleus RNA-seq (snRNA-seq). Since our review is a ‘Traditional Review’, the "method" section is not needed in the manuscript, so we are not added to the manuscript.

Point 5: A Flow chart diagram about the initial collected studies and the studies that finaly included in the rev should be added.

Response 5: Thank you for your advice. I am glad to learn a lot of new knowledge in your comments. This review is based on the transcriptomic studies of FFPE samples. The realization of snRNA-seq in FFPE samples requires two essential conditions: the preparation of nuclei and the construction of snRNA-seq technology platforms, the literature research in the early stage mainly focused on these two parts, and also involved some literature on sample protection. Thus, Flow chart diagram is not included in our review manuscript.

Point 6: What exclusion and inclusion criteria are the authors used to perform our review?

Response 6: Thank you very much for your constructive questions. Our review is mainly reported the nuclear preparation strategies of FFPE samples and snRNA-seq technologies, and does not involve medical evaluations, so there are no strict sample inclusion and exclusion criteria.

Round 2

Reviewer 5 Report

The revised version of the manuscript has significantly been improved.